# De Novo Design of Anti-COVID Drugs Using Machine Learning-Based Equivariant Diffusion Model Targeting the Spike Protein

Vidya Niranjan [1,*,†] , Akshay Uttarkar [1,†] , Ananya Ramakrishnan [1], Anagha Muralidharan [1], Abhay Shashidhara [1], Anushri Acharya [1], Avila Tarani [1] and Jitendra Kumar [2,*]

1   Department of Biotechnology, R V College of Engineering, Mysuru Road, Kengeri, Bangalore 560059, Karnataka, India
2   Bangalore Bioinnovation Centre (BBC), Helix Biotech Park, Electronics City Phase 1, Bengaluru 560100, Karnataka, India
*   Correspondence: vidya.n@rvce.edu.in (V.N.); director@bioinnovationcentre.com (J.K.)
†   These authors contributed equally to this work.

**Abstract:** The drug discovery and research for an anti-COVID-19 drug has been ongoing despite repurposed drugs in the market. Over time, these drugs were discontinued due to side effects. The search for effective drugs is still under process. The role of Machine Learning (ML) is critical in the search for novel drug compounds. In the current work, using the equivariant diffusion model, we built novel compounds targeting the spike protein of SARS-CoV-2. Using the ML models, 196 de novo compounds were generated which had no hits on any major chemical databases. These novel compounds fulfilled all the criteria of ADMET properties to be lead-like and drug-like compounds. Of the 196 compounds, 15 were docked with high confidence in the target. These compounds were further subjected to molecular docking, the best compound having an IUPAC name of (4aS,4bR,8aS,8bS)-4a,8a-dimethylbiphenylene-1,4,5,8(4aH,4bH,8aH,8bH)-tetraone and a binding score of −6.930 kcal/mol. The principal compound is labeled as CoECG-M1. Density Function Theory (DFT) and Quantum optimization was carried out along with the study of ADMET properties. This suggests that the compound has potential drug-like properties. The docked complex was further subjected to MD simulations, GBSA, and metadynamics simulations to gain insights into the stability of binding. The model can be in the future modified to improve the positive docking rate.

**Keywords:** equivariant diffusion model; inpaint; full-atom; SARS-CoV-2; molecular dynamics

## 1. Introduction

Coronavirus disease (COVID-19) is an infectious disease caused by SARS-CoV-2, which had an outbreak in December 2019 and has affected the mental, economical, and physical well-being of people across the globe. Symptoms of this include fever, shortness of breath, cough, and tiredness. Drugs include Metformin, which is the best line of medication for type 2 diabetes and other infectious diseases; although this drug does not have a proper pathway for preventing COVID-19, it improves the microbiome and has anti-inflammatory action [1]. Ivermectin is an antiparasitic drug and it helps in the inhibition of cytokine storm in COVID-19 [2]. Fluvoxamine is an antidepressant drug that after a few trials has raised the probability of reducing conditions such as hypoxemia in COVID-19 patients [3]. Although these drugs have been used in many clinical trials, they have failed to give effective results against COVID-19. SARS-CoV-2 contains 4 structural proteins (S, E, M, and N) and 16 non-structural proteins (nsp1−16) [4]. The heptad domain of the spike protein of COVID-19 is a promising target for the development of drugs and vaccines against the virus. This domain is involved in the process of fusion between the virus and the host cell, a crucial step in the infection cycle. Specifically, the heptad repeat region 1 (HR1) and

heptad repeat region 2 (HR2) of the spike protein form a six-helix bundle structure that facilitates fusion.

### 1.1. Diffusion Models for Molecules

This model uses non-equilibrium thermodynamics to understand data variation by modeling a reverse diffusion (denoising) process. An equivariant diffusion model is generated using DDPM (Denoising Diffusion Probabilistic Model) which is used for molecule generation in 3D. This model used Markov chains (a system that exhibits a change of state based on certain probabilistic rules) to produce samples within a finite time. These models are very easy to train and define, but they are not capable of producing good-quality samples. They also allow the use of a variable number of degrees of freedom. Using this, a high-quality image can be generated that can be used for further studies [5]. A diffusion model is proposed with a network that works on both continuous atomic coordinates and categorical atom types to produce new molecules in 3D space. For this method, the atoms need not be in a particular order and are more efficient than any other model. The probability of the formation of different molecules can be determined [6]. A type of diffusion that looks for a conditional setting where molecular conformations are developed from molecular graphs is called torsional diffusion. This process involves a framework that works on the space of torsion angles through a diffusion process of a hypertorus and an extrinsic to intrinsic model [7]. Hence, 3D diffusion models are used to design biomolecular structures such as proteins, antibodies, etc. The fundamental challenge faced in molecular science is Molecule design which is required for a variety of applications. It is practically not possible to perform tests and enumeration for all the different molecules available in wet labs. Thus, using the development of machine learning, models were generated. The molecule is usually represented as a string (1D), graph (2D), and geometry (3D). The existing design problems are according to problem setup which includes input, output, and goals [8].

### 1.2. Structure-Based Drug Design (SBDD)

Structure-based drug design involves finding ligand molecules that exhibit structure and chemical complementarity to protein pockets. Many methods have shown promise in proposing novel molecules from scratch [9]. In modern drug discovery, molecular docking methods are used to explore the complexes formed by the attachment of ligands in the binding sites of macromolecular targets [10]. Interactive software and human-based interaction are used so that candidate molecules bind with high affinity to the target. Using various docking algorithms, the pros and cons of any method are studied so that effective strategies can be developed for accurate results such as estimating the ligand-receptor binding free energy using calculating critical phenomena involving the intermolecular recognition process. The model is evaluated using a docking benchmark, and it is found that guided generation improves predicted affinities by drug-likeness by 10% over the baseline. In addition, the model proposes molecules with binding scores exceeding some known ligands, which find their application in wet-lab studies. The understanding of the 3D structure of biological molecules is performed either by using experimental methods or by predictions using the homology model. Two strategies were developed for the production of new ligands conditioned on protein pockets [11]. One of the strategies, protein-conditioned generation, was used to denoise molecules from fixed protein pockets, and the other strategy, ligand inpainting generation, approximated the distribution of ligand and pocket nodes. For sampling, the ligand was combined with a forward representation of the pocket in each denoising step [12].

Several approaches have been explored for designing drugs that target the heptad domain of the spike protein. One strategy is to develop small molecule inhibitors that interfere with the interaction between HR1 and HR2, thus preventing the formation of the six-helix bundle and inhibiting virus entry into host cells. Another approach is to

develop peptides or antibodies that mimic the structure of HR1 or HR2 and can bind to the complementary heptad region, disrupting the fusion process [13].

In this work, we aim at designing small molecules using an equivariant diffusion model targeting the heptad domain.

## 2. Materials and Methods

### 2.1. Selection of Spike Protein

The protein data bank (PDB) [14–16] consists of 1451 structures of COVID-19 with sources of electron microscopy (997), X-ray diffraction (443), and solution NMR (11) as of 15 November 2022. Based on the coverage, trimeric nature of the protein, and resolution of the structure, 6VSB [17,18] was selected to perform the de novo drug design. It is the prefusion model of spike protein with the ACE2 binding domain up.

### 2.2. Domain Analysis and Selection of Binding Sites

Prosite [19,20] and conserved domain database [21] were used to perform domain analysis and shortlist the receptor binding domain. The previously reported studies which include binding of prospective compounds were considered as a reference template to identify the binding sites.

### 2.3. Selection of Algorithm Model and Sampling Variables

We use an equivariant DDPM [22] to jointly create molecules and binding conformations regarding a given protein target. Protein and ligand point clouds are represented as fully connected graphs, which are then analyzed by EGNNs. We look at two methods of 3D pocket conditioning: (1) a conditional DDPM that gets a fixed pocket representation as a context in each denoising step; and (2) a model that approximates the joint distribution of ligand–pocket pairs using inpainting at inference time.

In the conditional full atom model and inpainting model, the samples were set to 100 and the nodes set to 20. The nodes were set based on the ACE2 receptor binding domain. Compounds with smaller nodes would reduce the "fitness" of the molecule in the binding pocket. Inversely, compounds with larger nodes would increase the cytotoxicity of the drug. The timesteps were set to 1000 to have a robust outcome.

For the inpainting model, resampling and jump length are set to 40 to have an optimal generation of de novo compounds.

### 2.4. ADME Properties and Toxicity Predictions

ADME and toxicity predictions were carried out using SwissADME [23] and Protox-II [24], free web-based tools. The range for each of the categories was obtained from in-build ADME tools in Schrodinger.

### 2.5. Molecular Docking

#### 2.5.1. Protein Preparation and Ligand Preparation

The Protein Preparation Workflow tool on Maestro Schrodinger 2022-3 [25] was used to fix issues with the protein and to maintain the pH of the environment besides determining ligands, metals, and co-solvents. To determine residues, HET atoms, and validate valency, the protein was pre-processed. During pre-processing, bond ordering was assigned, and hydrogens were replaced. Prime was utilized to compensate for the lack of side chains. Epik [26] was used to construct het states with pH values of $7.4 \pm 2.0$. During structural refining, PROPKA [27] was utilized to assign hydrogen bonds for neutral pH (7.0). Structure minimization for less than 0.30 was achieved using the OPLS3e [28] force field.

#### 2.5.2. Precision Docking

Glide [29–31] was used to perform molecular docking in extra precision docking mode. The scaling factor was set to 0.80 and the partial cut-off to 0.15. Ligand sampling was set to

flexible with bias sampling for torsions sets to predefined values of functional groups. No receptor constraints were set. The docked complexes were sorted based on the docking scores.

### 2.6. MD Simulation and Analysis

Desmond [32], a free academic users' software was used to perform the Molecular Dynamics simulation for a period of 100 ns. Two MD simulation runs were performed with the change in random seed value set to 2005 and 2010, respectively The interaction complex was subjected to protein pre-processing and H-bond assignment with similar parameters as mentioned earlier. The simulation system was built utilizing the system builder. The solvent model selected was TIP3P [33], with boundary conditions defined by the orthorhombic box with minimized volume encapsulating the complex. The force field applied is OPLS3e. The system was neutralized by adding 37 sodium ions at a 3.975 mM concentration. A total of 264,779 atoms were present in the system, comprising 71,758 water molecules. The detailed methodology for MD simulation studies can be found in our previous publications [34–37].

Simulation time was 100 ns, with trajectory recording interval set to 0.1 ns (the optimized interval time helps in clear visualization and appropriate storage space). Ensemble class was set to NPT (defines the thermodynamic parameters of constant pressure and temperature; a variable volume allows the protein to undergo conformational change). The complete simulation protocol involves the following steps:

1. Equilibration, Brownian Dynamics NVT, T = 10 K, small timesteps, and restraints on solute heavy atoms, 100 ps.
2. Equilibration, NVT, T = 10 K, small timesteps, and restraints on solute heavy atoms, 12 ps.
3. Equilibration, NVT and no restraints.
4. Final simulation (production MD) for 100 ns with time step was set to 2 femto-seconds (fs) and temperature of 303 K. A cut-off short range method with a radius of 9.0 Å (this optimized value avoids overlapping of atoms). The Noose-Hover chain thermostat method was used with RESPA integrator. Simulation has a temperature increase of 10 K per time step after solvation of the binding pocket.

### 2.7. MM-GBSA Analysis

The trajectory files from the MD simulations were used as the input for GBSA (Generalized Born and surface area solvation) calculations [38]. The trajectory consisting of 100 ns generated 10,000 frames. The structures were extracted every 10 frames, and a total of 10 structures were subjected to GBSA energy calculations.

VSGB 2.0 [39] solvation model was used for calculations. MM-GBSA generated a lot of energy properties. These properties reported energies for the ligand, receptor, and complex structures as well as energy differences relating to strain and binding, and they were broken down into contributions from various terms in the energy expression.

There were five fundamental energy calculations performed in Prime MM-GBSA: optimized free receptor (="Receptor"), optimized free ligand (="Ligand"), optimized complex (="Complex"), receptor of minimized/optimized complex, and the ligand of minimized/optimized complex.

From these energies, MM-GBSA dG Bind = Complex − Receptor − Ligand, and MM-GBSA dG Bind NS (No Strain) = Complex − Receptor (from optimized complex) − Ligand (from optimized complex). MM-GBSA dG Bind − Rec Strain − Lig Strain values are of crucial importance [40].

### 2.8. Metadynamics Simulation

Desmond's Metadynamics module was utilized to do the analysis. The factors that determined the simulation's accuracy were the height and breadth of the Gaussian potential, as well as the interval at which the Gaussians were introduced. The height-to-interval ratio had little influence on accuracy; nevertheless, lower values of this ratio somewhat

improved accuracy. During a free MD run, the Gaussian width should be roughly 1/4 to 1/3 of the collective variable's average fluctuations.

A CV wall equals the sum of the complex's largest dimensions. In this study, the wall was set at 100 Å as the protein–ligand overall system was 85.32 Å, which included the complete receptor–ligand complex.

The time interval between Gaussian injections was set at 0.09 picoseconds (ps). The simulation's temperature and pressure were set to 310 K and 1.01325 bar, respectively. The simulation time was set to 50 ns.

The complete protocol may be found in [41,42] and provides a method for calculating dissociation free energy (DFE) using free energy surface (FES) data.

### 2.9. Quantum Convergence and Single Point Energy Calculations

Jaguar [43] is a module that computes the electronic structure of molecules using ab initio quantum mechanics computations. The single point energy and quantum optimization of the principal compound were carried out.

Jaguar's reaction panel was used to configure the reaction energetics computation calculations based on Density function theory (DFT) and b3lyp-d3. In the molecules tab, symmetry was set to utilize if the charge was present, and multiplicity set to the molecular inherited attribute. The 6-31G [44] standard split valence double basis set was utilized, with polarization set to ** and diffuse set to none.

To execute a spin-unrestricted calculation for open-shell systems, the calculations were performed using density function theory with the SCF spin treatment set to automatic. It is critical to account for relativistic effects when dealing with heavy metals. Because the molecules lacked heavy metals, a non-relativistic approach was chosen.

For pseudo-spectral grids with a narrow cut-off, set the scheme to DIIS and the accuracy to ultrafine in SCF settings. Because the geometry of molecules was not refined at the atomic level, the original assumption was atomic overlap. In terms of convergence, the SCF level shift for Hartree with no thermal smearing was set at 0.0. The number of iterations was set to 100. The Poisson—Boltzmann Finite (PBF) model was chosen for solvation with water as a solvent because it yields higher energy than other PCM models. A detailed methodology is provided in publication [45].

Quantum optimization was performed before energy calculations. Molecular orbitals, atomic electrical potential charges, vibrational frequencies, and Raman spectroscopy property calculations were performed. Quantum convergence and Raman spectroscopy plots were generated.

### 3. Results

#### 3.1. Domain Analysis of Spike Protein

The spike protein is trimeric in nature. The protein information was obtained from the Uniport database [46] (Uniport ID P0DTC2) [47–49] with 1273 amino acids categorized into 16 domains. The domain of interest in our research objectives is to target the heptad repeats 1 and 2 to avoid interaction stability with human receptors [50].

Detailed information on the domain is provided in Table 1. The 3D structure of Spike protein is provided in Figure 1A.

#### 3.2. Novel Molecules Were Generated Using an Equivariant Diffusion Model

In total, 98 compounds were generated from inpainting and full atom model respectively. These 196 compounds generated in total are novel compounds with no hits in major chemical databases such as PubChem [51], ChemEMBL [52,53], ChemSpider, and many more.

Complete topological and physical descriptors of the compounds from the inpainting and full atom models are provided in Supplementary Files S1 and S2, respectively. The list of compounds in their 2D structures is provided in Supplementary Files S3 and S4, respectively.

The generated molecules fulfil the ADME and toxicity values required for drug-like compounds and are discussed in detail in a later section.

From each of the models used, 98 de novo compounds were generated. The compounds were subjected to fingerprinting based on atom pairs with 64-bit precision. Tanimoto similarity metrics were adapted as it is the widely used scheme. The linkage methods were set to average.

In the conditioned full atom model, 8 clusters were formed with a strain score of 1.104. A distance matrix in Figure 1B and a semi-partial R-squared plot in Figure 1D are provided. The clusters formed so subjected to molecular docking.

In the inpainting model, 3 clusters were formed with a cluster strain score of 1.109. A distance matrix in Figure 1C and a semi-partial R-squared plot in Figure 1E are provided.

**Table 1.** List of domains in spike region of SARS-CoV-2.

| Sl No | Amino Acids | Function |
|:---:|:---:|:---:|
| 1 | 280–301 | Putative superantigen; may bind T-cell receptor alpha/TRAC |
| 2 | 319–541 | Receptor-binding domain (RBD) |
| 3 | 403–405 | Integrin-binding motif; |
| 4 | 437–508 | Receptor-binding motif; binding to human ACE2 |
| 5 | 448–456 | Immunodominant HLA epitope recognized by the CD8+; called NF9 peptide |
| 6 | 681–684 | Putative superantigen; may bind T-cell receptor beta/TRBC1 |
| 7 | 816–837 | Fusion peptide 1 |
| 8 | 835–855 | Fusion peptide 2 |
| 9 | 920–970 | Heptad repeat 1 |
| 10 | 1163–1202 | Heptad repeat 2 |

*3.3. Molecular Docking and Interaction Pattern*

The compounds once clustered were subjected to molecular docking. The compounds from the inpainting model 10/98 compounds were docked showing energy value $< -2$ kcal/mol. The rest of the compounds had a positive dock score which is not thermodynamically viable binding energy.

Similarly, the compounds from the full atom model 5/98 were docked to the binding domain (heptad domain) of the spike protein. The success of binding action is very low compared to the number of compounds generated.

Out of the 196 compounds generated, 15 compounds were docked of which 6 compounds had good binding interaction in the heptad region. The interaction profile is provided in Figure 2A.

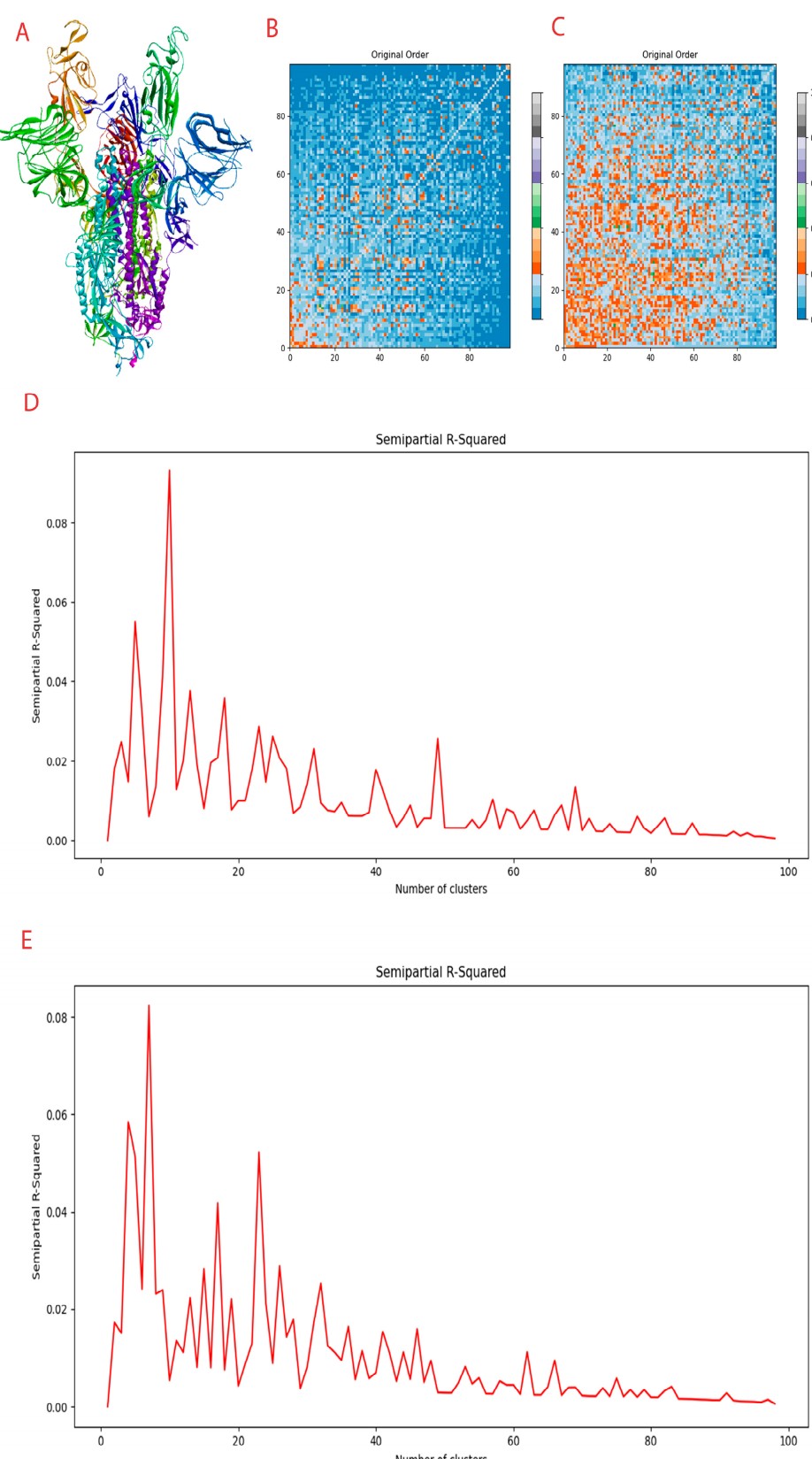

**Figure 1.** Compounds generated and clustering information. (**A**) Spike protein in the trimeric form. (**B**) Distance matrix of conditional full atom generated compounds. (**C**) Distance matrix of inpainting generated compounds. (**D**) Clustering R squared semi-empirical plot for conditional full atom plot. (**E**) Clustering R squared semi-empirical plot for inpainting plot.

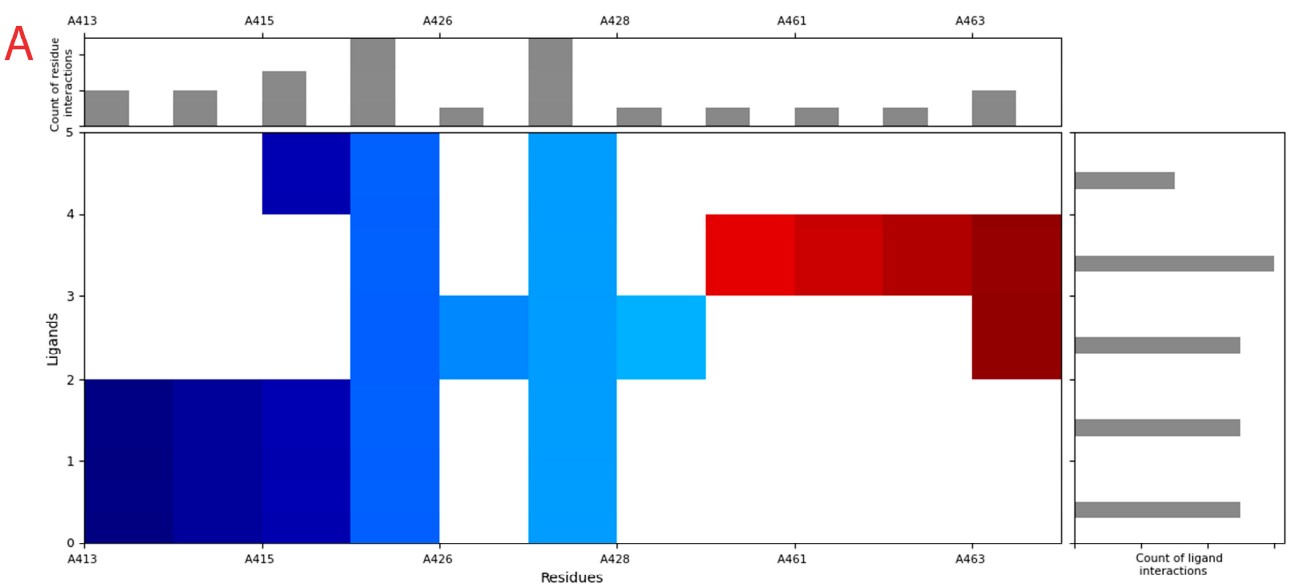

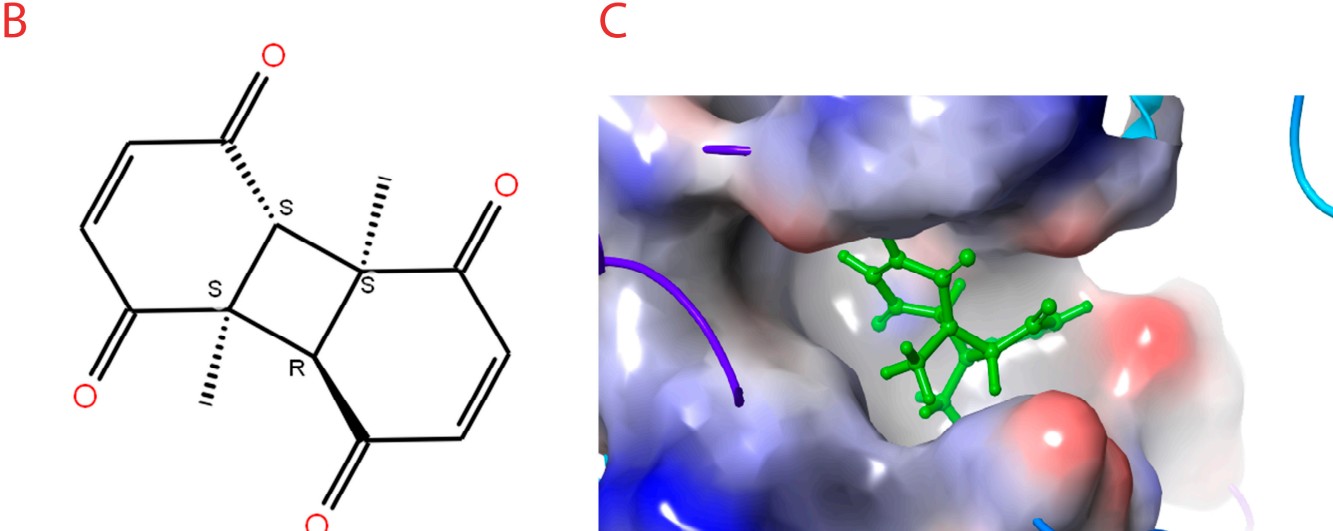

**Figure 2.** Interaction profile and details of CoECG-M1 interaction in depth. (**A**) The interaction profile of top compounds with the spike protein. The color code shifts are for the interaction from dark blue (left) to dark red (right) and are assigned to highlight each of the bars. (**B**) A Fischer structure of CoECG-M1. (**C**) The 3D interaction of CoECG-M1 with receptor residues. The binding surface is shown concerning the availability of acceptor or donor h-bond residues.

The compounds docked and their respective docking scores are provided in Table 2.

The compound with the best docking score was obtained using a full atom model bearing the SMILES O=C1C=CC(=O)[C@@H]([C@]12C)[C@]3(C)[C@H]2C(=O)C=CC3=O. The IPUAC name of the compound is (4aS,4bR,8aS,8bS)-4a,8a-dimethylbiphenylene-1,4,5,8(4aH,4bH,8aH,8bH)-tetraone. Henceforth, the compound will be addressed as the CoECG-M1. The Fischer representation of CoECG-M1 is Figure 2B. The molecule is reposited to PubChem bearing the SID 475724928 [54].

CoECG-M1 shows hydrogen bond interaction with Lys 1038, Arg 1107, and Asn 1108. Hydrogen bond interaction is the most effective mode of intra-molecular interaction. The docked complex along with the interaction diagram is shown in Figure 2C.

**Table 2.** List of compounds docked with heptad domain with compound number and representation in SMILES along with their docking score.

| Sl No. | Structure | Smiles | Docking Score in kcal/mol |
|--------|-----------|--------|---------------------------|
| 1 |  | O=C(O)C1CC(O)C(=O)C1(O)O | −4.346 |
| 2 |  | O=C(O)C1CCC(P(=O)(O)O)CCC1 | −4.192 |
| 3 |  | O=C1C(O)C(O)CC(C(=O)N1) | −4.112 |
| 4 |  | O=C1C(C(CC(N)C(N1)=O)O)O | −3.900 |
| 5 |  | O=C1C(O)C(O)CC(C(=O)N1)P(=O)(O)O | −3.842 |
| 6 |  | O=C(O)C(O)(O)C(C=O)CCOC(CC1)OC(O)C1O | −3.722 |

**Table 2.** *Cont.*

| Sl No. | Structure | Smiles | Docking Score in kcal/mol |
|---|---|---|---|
| 7 |  | OCCNC(=O)C1CC(P(=O)(O)O)OC(=O)C1=O | −3.616 |
| 8 |  | O=C(O)C(=O)NCCCO | −3.508 |
| 9 |  | O=C(O)N1CCN(C1O)C(=O)C(O2)CC(C23)CCC3=O | −3.503 |
| 10 |  | N=C(N1CCN(C(C2OC3C(CCC3=O)C2)=O)C1O)O | −3.438 |
| 11 |  | O=C(O)C(O)(O)C(C=O)CCOC(CC1)OC(O)C1O | −3.234 |

**Table 2.** *Cont.*

| Sl No. | Structure | Smiles | Docking Score in kcal/mol |
|---|---|---|---|
| 12 | | O=C(N1CCN(C(C2NC3C(CCC3=O)N2)=O)C1=O)O | −2.676 |
| 13 | | O=C(O)C1CCC(P(=O)(O)O)CCC1 | −2.225 |
| 14 | | O=C(O)N1CCN)C(O2)CC(C23)CCC3=O | −2.211 |
| 15 | | O=C1C=CC(=O)[C@@H]([C@]12C)[C@]3(C)[C@H]2C(=O)C=CC3=O | −6.930 |

*3.4. CoECG-M1 Shows Potential Drug-like Ability via ADMET Predictions*

This section describes various properties and characteristics of a solute molecule, including its molecular weight, dipole moment, solute-accessible surface area (SASA), hydrophobicity, ionization potential, electron affinity, predicted polarizability, logP, solubility, serum protein binding, brain/blood partition coefficient, number of primary metabolites, CNS activity, the potential for blocking the HERG K+ channel, permeability through Caco-2 and MDCK cell lines, skin permeability, transdermal transport rate, compliance with Lipinski and Jorgensen rules, oral absorption, and qualitative model for oral absorption. Additionally, the report lists the names and similarity scores of four molecules that are most like CoECG-M1. Table 3 contains information regarding the values, range for each of the above mentioned property.

**Table 3.** List of ADME properties along with its range values concerning 95% of FDA-approved drugs.

| Sl No. | Property | Value | Range (95% of Drugs) |
|--------|----------|-------|----------------------|
| 1 | Molecular Weight | 244.246 | 130.0/725.0 |
| 2 | Dipole Moment | 1.678 | 1.0/12.5 |
| 3 | Total SASA | 424.755 | 300.0/1000.0 |
| 4 | Hydrophobic SASA | 125.120 | 0.0/750.0 |
| 5 | Hydrophilic SASA | 154.362 | 7.0/330.0 |
| 6 | Carbon Pi SASA | 145.273 | 0.0/450.0 |
| 7 | Weakly Polar SASA | 0.000 | 0.0/175.0 |
| 8 | Molecular Volume | 747.138 | 500.0/2000.0 |
| 9 | vdW Polar SA | 101.833 | 7.0/200.0 |
| 10 | Rotatable Bonds | 0.000 | 0.0/15.0 |
| 11 | Hydrogen Bonds | 0.000 (Donor)/8.000 (Acceptor) | 0.0/6.0 (Donor)/2.0/20.0 (Acceptor) |
| 12 | Globularity | 0.937 | 0.75/0.95 |
| 13 | Ionization Potential | 10.847 (eV) | 7.9/10.5 |
| 14 | Electron Affinity | 1.316 (eV) | $-0.9/1.7$ |
| 15 | QP Polarizability | 25.134M ($\text{Å}^3$) | 13.0/70.0 |
| 16 | QP log P (hexadecane/gas) | 7.710M | 4.0/18.0 |
| 17 | QP log P (octanol/gas) | 12.663 | 8.0/35.0 |
| 18 | QP log P (water/gas) | 10.659 | 4.0/45.0 |
| 19 | QP log P (octanol/water) | $-0.625$ | $-2.0/6.5$ |
| 20 | QP log S (aqueous solubility) | $-0.078$ | $-6.5/0.5$ |
| 21 | QP log S (conformation independent) | $-0.770$ | $-6.5/0.5$ |
| 22 | QP log K (serum protein binding) | $-1.353$ | $-1.5/1.5$ |
| 23 | Caco-2 Permeability (nm/s) | 340 | <25 poor, >500 great |

Primary Metabolites and Reactive FGs can be described as metabolism likely as that of alpha hydroxylation of carbonyl.

The other important parameters under consideration are as follows:

1.  Lipinski Rule of 5 Violations = 0 (maximum is 4)
2.  Jorgensen Rule of 3 Violations = 0 (maximum is 3)
3.  % Human Oral Absorption in GI (+−20%) = 69 (<25% is poor)
4.  Qual. Model for Human Oral Absorption = Medium (>80% is high)
5.  CoECG-M1 has percentage similarity with Chlormezanone [55] (83.97%), Meticrane [56] (80.06%) Pemirolast [57] (77.62%), and Aceglatone [58] (76.98%)

Toxicity prediction suggests the $LD_{50}$ value of 2300 mg/kg and it belongs to class 5. Details of other parameters are provided in Table 4.

**Table 4.** Organ, tissue, and pathways-specific toxicity profile along with prediction probability values.

| Sl No. | Classification | Target | Prediction | Probability |
|---|---|---|---|---|
| 1 | Organ toxicity | Hepatotoxicity | Inactive | 0.75 |
| 2 | Toxicity endpoints | Carcinogenicity | Inactive | 0.51 |
| 3 | Toxicity endpoints | Immunotoxicity | Inactive | 0.57 |
| 4 | Toxicity endpoints | Mutagenicity | Inactive | 0.62 |
| 5 | Toxicity endpoints | Cytotoxicity | Inactive | 0.79 |
| 6 | Tox21-Nuclear receptor signalling pathways | Aryl hydrocarbon Receptor (AhR) | Inactive | 0.84 |
| 7 | Tox21-Nuclear receptor signalling pathways | Androgen Receptor (AR) | Inactive | 0.93 |
| 8 | Tox21-Nuclear receptor signalling pathways | Androgen Receptor Ligand Binding Domain (AR-LBD) | Inactive | 0.98 |
| 9 | Tox21-Nuclear receptor signalling pathways | Aromatase | Inactive | 0.87 |
| 10 | Tox21-Nuclear receptor signalling pathways | Estrogen Receptor Alpha (ER) | Inactive | 0.68 |
| 11 | Tox21-Nuclear receptor signalling pathways | Estrogen Receptor Ligand Binding Domain (ER-LBD) | Inactive | 0.85 |
| 12 | Tox21-Nuclear receptor signalling pathways | Peroxisome Proliferator-Activated Receptor Gamma (PPAR-Gamma) | Inactive | 0.97 |
| 13 | Tox21-Stress response pathways | Nuclear factor (erythroid-derived 2)-like 2/antioxidant responsive element (nrf2/ARE) | Inactive | 0.90 |
| 14 | Tox21-Stress response pathways | Heat shock factor response element (HSE) | Inactive | 0.90 |
| 15 | Tox21-Stress response pathways | Mitochondrial Membrane Potential (MMP) | Inactive | 0.57 |
| 16 | Tox21-Stress response pathways | Phosphoprotein (Tumour Suppressor) p53 | Inactive | 0.77 |
| 17 | Tox21-Stress response pathways | ATPase family AAA domain-containing protein 5 (ATAD5) | Inactive | 0.90 |

### 3.5. A Comprehensive Simulation Studies Provide Insights into Stability, Binding and Unbinding Energies of CoECG-M1

The lead compound (CoECG-M1) was subjected to molecular dynamics simulation to provide insights into the stability of binding efficiency of the lead compound. Analyses were performed for protein–ligand RMSD, protein–ligand RMSF, protein–ligand interaction profile over the simulation period, and radius of gyration. The details of the analyses are provided in the following section.

3.5.1. Molecular Dynamic Simulation for 100 ns

The docked complex of CoECG-M1 with spike protein was subjected to MD simulation studies for 100 ns. The plot (Fig) shows the Root Mean Square Deviation (RMSD) evolution of a protein (left Y-axis). All protein frames are first aligned on the reference frame backbone, and then the RMSD is calculated based on the C-alpha atom selection. The frame 0 is considered as reference for RMSD calculations. Monitoring the RMSD of the protein can give insights into its structural conformation throughout the simulation. The protein RMSD is observed to be at 5.8 and 7.2 angstroms in the simulation 1 and simulation 2 runs performed, respectively, for 100 ns each. The other parameters for comparison of results between MD run 1 and MD run 2 are provided in Table 5. For small globular proteins, it is expected to be

in the range of 3 to 6 angstroms [59]. This RMSD is provided in Figure 3A. CoECG-M1 is firmly bound to the binding pocket. The Root Mean Square Fluctuation (RMSF) in Figure 3B is useful for characterizing local changes along the protein chain. Peaks indicate areas of the protein that fluctuate the most during the simulation. Typically, you will observe that the tails (N- and C-terminal) fluctuate more than any other part of the protein. Secondary structure elements such as alpha helices and beta strands are usually more rigid than the unstructured part of the protein and thus fluctuate less than the loop regions.

**Table 5.** Comparison of MD simulation results for replica runs.

| Sl No. | Measure | Run 1 | Run 2 |
|--------|---------|-------|-------|
| 1 | Protein RMSD | 5.8 angstroms | 7.2 angstroms |
| 2 | Ligand RMSD | 0.35 angstrom | 0.28 angstrom |
| 3 | Secondary structure elements | 37.8% | 38.7% |
| 4 | H-Bond interactions | Lys 1038, Arg 1107 and Asn 1108 | Arg 1107 and Asn 1108 |
| 5 | Radius of gyration | 2.64 angstroms | 2.64 angstroms |

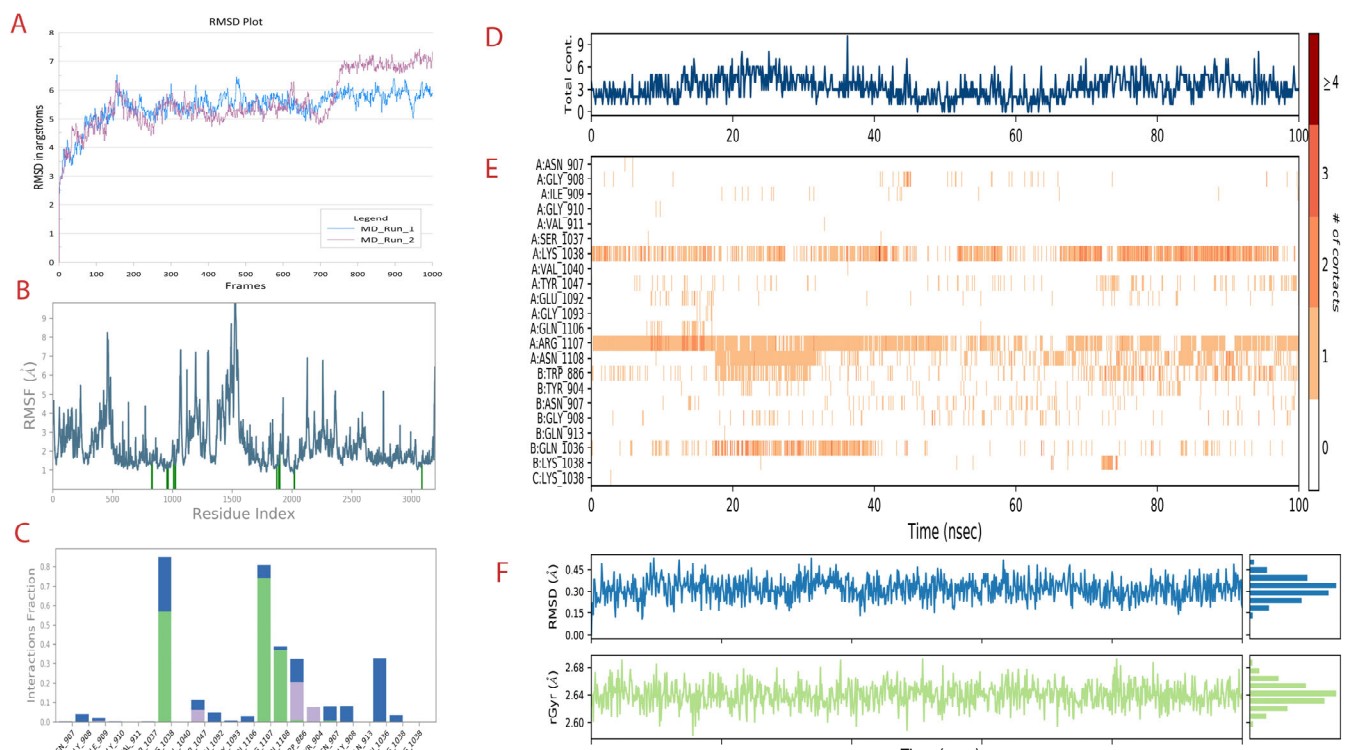

**Figure 3.** MD simulation run 1 analysis of the spike-CoECG-M1 compound. (**A**) RMSD plot of spike protein for both the MD runs performed shown in blue and purple. (**B**) The RMSF plot of fluctuations of residues with the interaction with CoECG-M1; the blue lines highlight the interacting amino acids. (**C**) The ligand–receptor contact plot was recorded for the simulation time. Green bars are for hydrogen bonds, blue for water bridges, and grey for hydrophobic interactions. (**D**) Ligand contact map with total contacts versus simulation time. (**E**) Extension of Figure 4D with details of contact specific to residues over the simulation period in nanoseconds. # refers to number of contacts from 0 to >4 as per legend provided. (**F**) RMSD and radius of gyration for the principal compound (CoECG-M1) representing the overall "fitness".

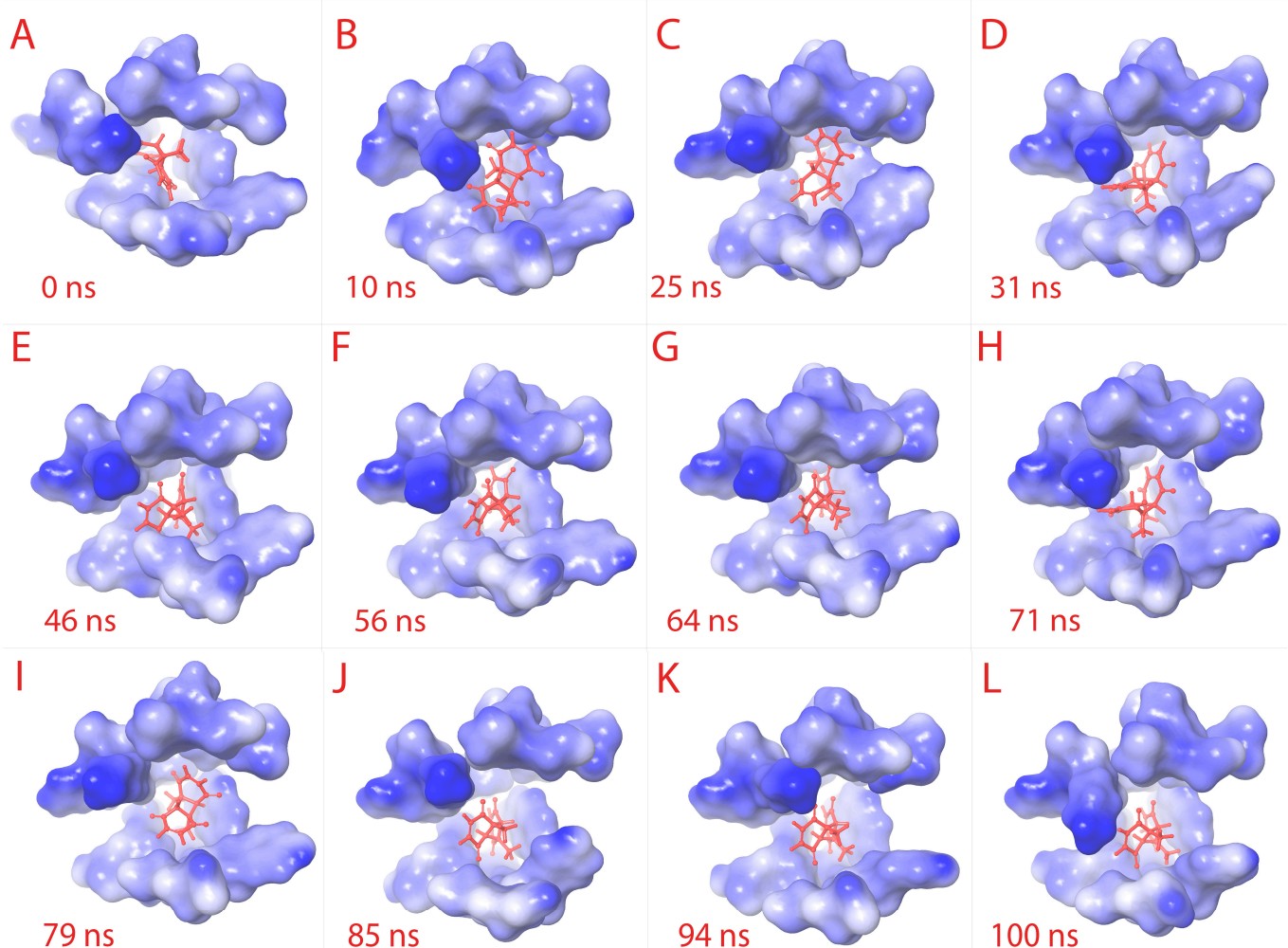

**Figure 4.** Ligand RMSD visualization within the binding pocket. The ligand (red) binding site amino acids structures are retrieved to visualize the RMSD of ligand. From (**A–L**) sequentially, the binding pocket is color-gradient based on the electrostatic potential (blue is negative, and white is positive). The respective time frame of capture is shown for each sub-figure.

Protein interactions with the ligand can be monitored throughout the simulation. The interactions can be categorized as h-bonds, hydrophobic, ionic, and water bridges.

Over the simulation period, h-bond interactions are seen with Lys 1038, Arg 1107, and Asn 1108. This has been provided in Figure 3C. Since the simulation was implicit by nature, the presence of water molecules mediated hydrogen-bonded protein–ligand interactions. All the h-bond interactions are water-mediated interactions. Ionic interactions are between two oppositely charged atoms that are within 3.7 Å of each other and do not involve a hydrogen bond.

The contact map provided in Figure 3D shows that there have been constant bonds and interactions throughout the simulation time with the receptor.

Overall, constant interaction was occurring with Lys 1038 and Arg 1107. Partial but more than sufficient interaction was observed with Asn 1108. Other amino acids which are part of the heptad domain interactions were observed in "on and off" mode which can be visualized in Figure 3E.

Since the compound is novel in nature, understanding the compound RMSD and radium of gyration is necessary to gain insights into the binding stability. For ligand RMS, out of 1000 frames from MD simulation, apart from the 0th frame and 1000th frame, 10 intermediate frames were randomly selected to calculate the RMS values. The change

in ligand conformation and orientation is found to be 0.283 angstroms with the standard deviation of 0.08 angstroms. The ligand lacks any torsion bonds and, hence, the possibility of it undergoing any conformational changes is low. The ligand is found to be in the binding pocket throughout the simulation run.

The radius of gyration is a measure of the distribution of mass about an axis of rotation in a rigid body. It is defined as the square root of the ratio of the moment of inertia of a body to its mass. The smaller the radius of gyration is, the more compact the body's mass is, and the greater its rotational stability is. For the compound in the study, the value is 2.64 angstrom.

Both of the results together suggest that the "fitness" of the compound within the receptor pocket is efficient. The plots are provided in Figure 3F. From Figure 4A–L, the binding pocket along with the ligand is captured at different time frames. The simulation video is provided in Supplementary File S5.

### 3.5.2. Molecular Mechanics with Generalized BORN and Surface area Solvation (MM/GBSA) Studies

MMGBSA calculations [60] were performed for accurate binding energy of CoECG-M1 with spike protein. The 100 ns simulation time was split into 10 frames of 10 ns each, and calculations were formed for each frame. The detailed value of each frame is provided in Table 5. The DG bind energy was found to be −41.16 kcal/mol (+/−) 3.68 kcal/mol for MD run 1 and −38.79 kcal/mol (+/−) 3.36 kcal/mol for MD run 2, and similarly the DG bind (no strain) energy was −42.13 kcal/mol (+/−) 3.81 kcal/mol for MD run 2 and −39.47 kcal/mol (+/−) 3.47 kcal/mol. The of MMGBSA values for each frame at regular time intervals has been provided in Table 6.

**Table 6.** Comprehensive list of binding energy values calculated via MM/GBSA method for every 10 nanoseconds.

| Time (ns) | MD Run 1 | | MD Run 2 | |
|---|---|---|---|---|
| | DG Bind (kcal/mol) | DG Bind NS (kcal/mol) | DG Bind (kcal/mol) | DG Bind NS (kcal/mol) |
| 10 | −38.37404477 | −40.6192763 | −39.51735788 | −39.82294397 |
| 20 | −38.42947824 | −39.1927792 | −41.90389158 | −43.09712232 |
| 30 | −42.21866335 | −42.59491568 | −37.4298756 | −37.95905421 |
| 40 | −41.55274598 | −42.46953731 | −47.63490585 | −48.53847099 |
| 50 | −41.29063169 | −41.50944017 | −36.60442487 | −37.80706116 |
| 60 | −39.52055104 | −40.4680222 | −36.70081353 | −37.67308507 |
| 70 | −35.63802159 | −36.09386049 | −36.82613064 | −37.68762383 |
| 80 | −41.09375804 | −42.18120988 | −37.641591 | −37.99902137 |
| 90 | −48.9177109 | −50.1835614 | −37.55085127 | −37.15958535 |
| 100 | −44.66043935 | −46.04530244 | −36.10956839 | −36.99196607 |

### 3.5.3. Meta Dynamics Studies to Gain Insights on the Unbinding Potential of CoECG-M1

To calculate the unbinding energy, we used meta dynamics simulation [61]. The protein–ligand unbinding energy is the measure of the energy required for the protein and ligand to separate from one another. It is usually measured in kcal/mol and is a useful indicator of the strength of the protein–ligand interaction. The −12.10 kcal/mol unbinding energy and −41.16 kcal/mol unbinding energy for protein–ligand is considered to be feasible, as both values fall within the expected range. This suggests that the interaction between the protein and ligand is strong enough to remain stable in physiological conditions. The free energy surface plot is provided in Figure 5A.

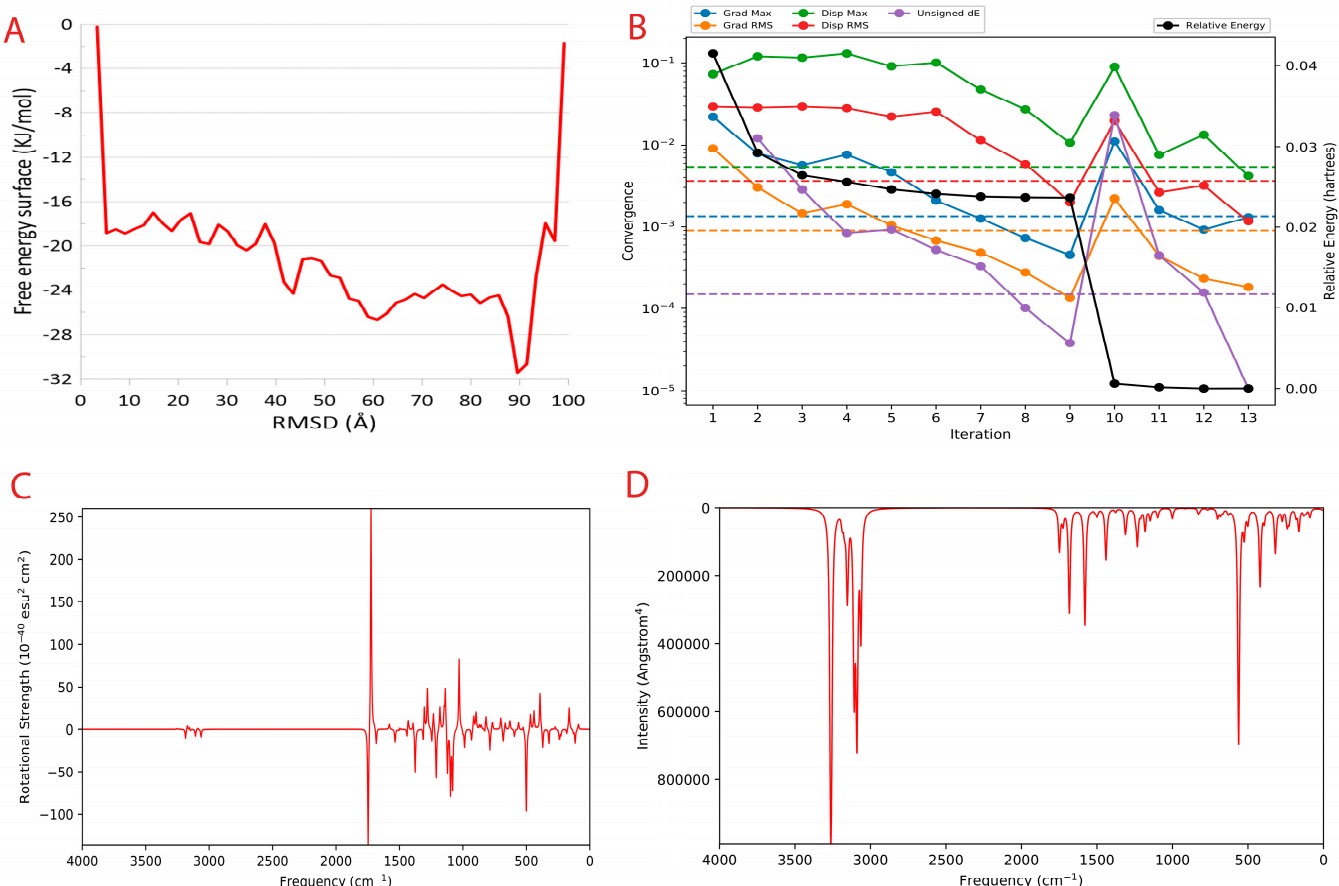

**Figure 5.** Metadynamics and Quantum computing analysis. (**A**) The free energy surface (FES) plot obtained from the metadynamics simulation. (**B**) The quantum convergence plot for the principal compound (CoECG-M1) runs with convergence and relative energy against a number of iterations to achieve the same. (**C**) Vibrational Circular Dichroism (VCD) plot for the principal compound. (**D**) Raman spectroscopy plot for the principal compound.

### 3.6. Quantum Insight into the Stability, Reactivity, and Structural Characteristics of the Principal Compound

There were 330 canonical orbitals used in the calculation, and the energy of the system was determined both in the gas phase and the solution phase.

Quantum convergence via optimization was carried out before the calculation of energy; the 0.00 Hartree was achieved after 13 iterations. The convergence plot is provided in Figure 5B.

The final energy of the system was found to be −841.592671 Hartree. The solvation energy of the system was −14.83 kcal/mol. The highest occupied molecular orbital (HOMO) and the lowest unoccupied molecular orbital (LUMO) were determined to be −0.258569 Hartree and −0.113911 Hartree, respectively. The lowest, highest, and second-lowest vibrational frequencies of the system were −199.41 $cm^{-1}$, 3264.197 $cm^{-1}$, and −150.064 $cm^{-1}$, respectively. The zero-point energy of the system was found to be 144.19 kcal/mol. The entropy, enthalpy, and free energy of the system at 298.15 K and $1.00 \times 10^{+00}$ atm were 111.977 cal/mol/K, 8.988081 kcal/mol, and −24.397935 kcal/mol, respectively. The internal energy and heat capacity of the system was 8.395596 kcal/mol and 55.018 cal/mol/K, respectively. The natural logarithm of the partition function was 41.179. Finally, the total internal energy, enthalpy, and free energy of the system in atomic units (au) at 298.15 K and $1.00 \times 10^{+00}$ atm were found to be −841.334582 au, −841.333637 au, and −841.386841 au, respectively.

Vibrational Circular Dichroism (VCD) is a spectroscopic technique used to measure the chirality of small molecules. It exploits the differences in the way right- and left-handed

molecules absorb left- and right-circularly polarized infrared light. It is an attractive alternative to other chiral techniques such as X-ray crystallography and Nuclear Magnetic Resonance (NMR), as it is a non-destructive technique that can be used on molecules in solution. VCD is a powerful tool for characterizing small molecules and can be used to help design novel compounds with desired properties. The VCD plot is provided in Figure 5C.

Raman spectroscopy is a powerful tool for analyzing the vibrational and rotational structure of small molecules. It can be used to identify novel small molecules and to determine the structure of new compounds. Raman spectroscopy works by using light to excite the molecules in a sample and then measure the light that is scattered by the sample. This scattered light contains information about the vibrational and rotational structure of the molecules in the sample. Raman spectroscopy can be used to determine the structure of small molecules, as well as to identify new and unknown molecules. The Raman spectra plot CoECG-M1 is provided in Figure 5D.

## 4. Discussion

In the recent studies and reviews, a lot of concerns have been raised on the ineffectiveness of the repurposing of drugs [62–65], which can also be due to mutations [66] in target proteins. This has led to the search for new active sites on receptors. The heptad repeat domain (HRD) on the spike protein of SARS-CoV-2 (severe acute respiratory syndrome coronavirus 2) is a potential drug target for treating COVID-19 [13]. The spike protein plays a crucial role in the virus's ability to infect host cells, and the HRD is an essential component of the protein's function [67].

The HRD is a sequence of seven amino acids repeated multiple times within the spike protein. It is involved in the formation of a stable, trimeric structure that allows the spike protein to bind to the host cell's angiotensin-converting enzyme 2 (ACE2) receptor [68]. This binding initiates the fusion of the virus's and host cell's membranes, which is a crucial step in the virus's replication cycle. Several studies have shown that drugs targeting HRD could potentially block the virus's ability to infect host cells. One class of drugs being explored is the antiviral agents that target the HRD's hydrophobic pockets. These agents are designed to bind to the HRD and disrupt the formation of the trimeric spike protein structure, preventing the virus from binding to the ACE2 receptor. Another approach is the development of antibodies that target HRD [69]. These antibodies could bind to the HRD and prevent the spike protein from forming its trimeric structure, effectively blocking the virus's ability to infect host cells. Monoclonal antibodies (mAbs) are a type of antibody that is engineered to bind specifically to a target, such as the HRD on the spike protein. Several mAbs targeting HRD have been developed and are currently in clinical trials [70].

There are also small-molecule drugs being developed that target HRD [71]. These drugs are designed to bind to the HRD and disrupt its function, preventing the virus from infecting host cells. Some of these drugs are designed to mimic the structure of the HRD and bind to the same hydrophobic pockets that the antiviral agents target, while others are designed to bind to specific amino acids within the HRD sequence.

Itraconazole [72,73] and Estradiol benzoate [74] have been found to inhibit viral entry by targeting the six-helix fusion core of SARS-CoV-2 S protein. These drugs were identified as potential treatments for COVID-19 through further studies that showed they can interact with the heptad repeat 1 (HR1) region of the spike protein. This is a promising development in the search for effective treatments against the virus, as there are currently no clinically effective drugs against SARS-CoV-2 infection. However, it should be noted that further research is needed to determine the safety and efficacy of these drugs in treating COVID-19 [75].

The HRD is an attractive target for several reasons. First, it is a highly conserved domain within the spike protein, which means that it is unlikely that the virus will evolve to become resistant to drugs targeting HRD. Second, the HRD is involved in a critical step in the virus's replication cycle, so drugs targeting the HRD have the potential to be highly effective at blocking the virus's ability to infect host cells.

Despite the potential of HRD as a drug target, there are also some challenges to developing drugs that target HRD. One challenge is that the HRD is a small, highly flexible domain, which makes it difficult to design drugs that bind specifically to the HRD. Additionally, the HRD is involved in several other functions within the spike protein, so drugs targeting the HRD must be carefully designed to avoid disrupting these functions.

There are several anti-heptad peptides and monoclonal antibodies designed against them, but no work has been conducted on drug design. The list is provided below.

1. Bamlanivimab: a monoclonal antibody that targets the HR1 domain of the spike protein and blocks its interaction with the human ACE2 receptor [76].
2. Casirivimab and Imdevimab: a combination of two monoclonal antibodies that target the HR1 and HR2 domains of the spike protein [77].
3. VIR-7831: a monoclonal antibody that targets the HR1 domain of the spike protein [78].
4. REGN-COV2: a combination of two monoclonal antibodies that target different domains of the spike protein [79].
5. Fc-engineered antibodies: a class of antibodies that have been modified to enhance their ability to neutralize the virus by targeting the HR1 and HR2 domains of the spike protein [80].

These drugs are designed to neutralize the virus by blocking its ability to bind to human cells, thereby preventing it from entering and replicating within the host. They are being tested in clinical trials for their efficacy and safety as treatments for COVID-19, and some have already been granted emergency use authorization by regulatory agencies for their use in certain populations.

It is important to note that the development of these drugs is an ongoing process, and the list is subject to change as more information becomes available. Additionally, it is also important to consider that while these drugs may be effective at blocking the virus, they are not a cure for COVID-19 and must be used in combination with other treatments, such as supportive care, to provide optimal results.

Machine learning-based diffusion models are a promising approach in the design of novel anti-COVID-19 drugs. The COVID-19 pandemic has led to an unprecedented global effort to develop effective treatments and vaccines, and machine learning has emerged as a key tool in this effort. Machine learning models are used to analyze high-throughput screening data from large libraries of compounds, such as those in pharmaceutical company collections or publicly available sources. This allows researchers to identify compounds that exhibit the desired properties, such as binding to specific targets or inducing specific cellular responses, and prioritize them for further development.

Additionally, machine learning-based diffusion models can also be used to analyze the results of in vitro assays, such as cell-based assays or biochemical assays. This can help identify compounds that exhibit the desired biological activity and predict their efficacy in vivo, based on their physical properties and the properties of the target molecules. One key advantage of machine learning-based diffusion models is their ability to handle large amounts of data and to identify complex relationships between variables. This makes them particularly useful for identifying potential anti-COVID-19 drugs, as the virus has many different targets, and it is challenging to predict how drugs will interact with these targets.

However, there are also some limitations to machine learning-based diffusion models. One challenge is the quality and quantity of data available for the training and validation of the models. In the current case, only 15/196 molecules were docked efficiently. Additionally, the models may be limited by the complexity of the biological systems they aim to simulate.

Despite these limitations, machine learning-based diffusion models have already shown promise in the design of novel anti-COVID-19 drugs. By providing a systematic and data-driven approach to drug design, these models have the potential to accelerate the discovery and development of effective treatments for COVID-19.

## 5. Conclusions

Machine learning models, efficient implementation, and their application are of great importance in the current research era of computational biology. All the 196 compounds predicted were not available in any of the chemical databases including PubChem, ChemEMBL, and Chem Spider. The design of such novel drug molecules under structure-based drug design will be pivotal in overcoming the limitations of traditional drug design approaches.

The designed anti-COVID-19 drug (CoECG-M1) has shown impressive in silico properties to be a potential lead drug candidate. In the future, in vitro and in vivo studies will be carried out.

The major limitation of the current model is the inability to assign appropriate bond orders and fulfil the valency criteria of the compounds. Similarly, the bond length and dihedrals are not accurate, and, hence, minimization of compounds must be performed before further processing. The rate of compounds with a good docking score compared to predicted positive compounds is very low. This is due to the lack of incorporation of charges of residues on the receptor binding pocket.

In the future, the charges of the receptor pocket need to be taken into consideration to increase the success rate.

**Supplementary Materials:** The following supporting information can be downloaded at: https://www.mdpi.com/article/10.3390/cimb45050271/s1. Supplementary File S1: Molecular and topological descriptors for the compounds generated using the inpaint model. Supplementary File S2: Molecular and topological descriptors for the compounds generated using full atom model. Supplementary File S3: 2D structures of compounds generated using inpaint model. Supplementary File S4: 2D structures of compounds generated using full atom model. Supplementary File S5: A video of the MD simulation performed for 100 ns with Spike protein chain A, B and C in green, magenta and cyan color respectively. The binding pocket is shown in surface representation colored in yellow and the ligand (CoECG-M1 highlighted in red).

**Author Contributions:** V.N. was in for ideation and conceptualization. A.U. contributed to the overall methodology, DFT (Density Function Theory), optimization, and metadynamics. A.R. and A.M. were involved in the design of molecules. A.S., A.A. and A.T. worked on docking and MD. simulation. A.R., A.M., A.S., A.A., A.T. and A.U. were involved in drafting the manuscript. J.K. was involved in refining the methodology and funding acquisition. All authors have read and agreed to the published version of the manuscript.

**Funding:** The funding was acquired from the Bangalore Bioinnovation Centre; the Karnataka Innovation and Technology Society; the Department of Electronics, IT, BT, and S&T; and the Government of Karnataka, India, and was put towards paying the publication cost.

**Institutional Review Board Statement:** Not applicable.

**Informed Consent Statement:** Not applicable.

**Data Availability Statement:** The protein structure used in the analysis is available in the PDB database with PDB ID 6VSB (http://doi.org/10.2210/pdb6VSB/pdb) (accessed on 14 October 2022). The principal compound (CoECG-M1) has been uploaded in PubChem, and it is available with SID 475724928.

**Acknowledgments:** The authors acknowledge the Bangalore Bioinnovation Centre; the Department of Electronics, IT, BT, and S&T; and the Government of Karnataka, India, for funding, which was put towards paying the publication cost. The authors thank the Department of Computer Science and Engineering, the R V College of Engineering, and Bangalore for providing GPU (NVIDIA A100) computational support. A warm heartfelt thanks to the staff and administration at the R V College of Engineering for their support.

**Conflicts of Interest:** The authors declare no conflict of interest.

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
