# Peer review of "De Novo Design of Anti-COVID Drugs Using Machine Learning-Based Equivariant Diffusion Model Targeting the Spike Protein"

_cimb, doi:10.3390/cimb45050271_

Round 1

Reviewer 1 Report

The manuscript is poorly organized overall, and it needs major revisions in order to be of acceptable quality; I will briefly underline the major ones for the authors.

Mos importantly, you wrote lengthy introductions and Discussion with general topics, but provided an etremely poor description of the process leading to the selection of 98 compounds selected using machine learning methods, to the further selection of 10 among them for studies, and for focusing on the selected lead (apart from "best docking results). Such process must be well explained, and decisions must be substantiated.

A two pages' discussion only talks abot general topics (i.e., machine learning, structural features of spike and known modulators), but does not even mention the selected lead! It should be centered on your findings in the project, not on literature. Similarly to revise and focus is the short Conclusions' paragraph

Overall, English is average, but several points / paragraphs should be rechecked carefully by a mother English language speaker.

I do find merit and relevance in the proposed work, but a major revision needs to be carried out and to be reviewed before accepting the manuscript.

Author Response

We request the reviewer to please see the attachment

Reviewer 2 Report

The manuscript entitled “Denovo design of anti-COVID drugs using machine learning based equivariate diffusion model targeting the spike protein” relies on computational studies of new compounds for COVID-19 treatment. Although, the idea of the manuscript about searching new compounds is good and could fit with the journal scope; the manuscript presents many conceptual errors, the introduction is poor presented and there are several English and grammatical issues. The results are poor presented, machine learning methodology and results found are not well described (as tittle suggests) and discussion is missing.

1. The manuscript must be revised by an English language expert.

2. Authors sometimes call the disease COVID-19, covid, covid 19, etc. There is no homogeneity in the text. Also for SARS-CoV-2 that sometimes is called SARS-nCoV-2 and do not understand why.

3. Authors explain in page 2 line 92-93

“The domain of interest in our research objectives is to target the heptad repeats 1 and 2 to avoid interaction stability with human receptors.”

But in the results, they said that they perform molecular docking calculations in the RBD subunit, page 4, line 125-126.

“Similarly, for the compounds from the full atom model 5/98 were docked to the receptor binding domain of the spike protein.”

4. What is a binding action? Line 126, conceptual mistake

5. Table 2, docking scores does not have units.

6. Page 5, lines 141-142. Authors stated:

“CoECG-M1 shows pi-pi interaction with Lys 1038, Arg 1107 and Asn 1108. Pi-pi in- teraction is most effective mode of intra-molecular interaction leading by hydrogen bond.”

Are they saying that pi-pi interaction lead to H-bond interaction. It does not have sense. Also, Lys, Arg and Asn cannot form pi-pi interactions that are related to Trp, Tyr or Phe amino acids, because they involve (poly)cyclic molecules.

7. Figures must be improved and well described. For exampled, Figure 2A what do colors red and blue means? What is the y axes ligands? And, count of ligand interactions does not have numbers or units. It is not clear the information in that presentation.

8. Concerning MD simulations, what do you use as reference frame for RMSD calculations? Frame 0? Crystallographic structure? Optimized structure?

9. Line 185, authors said “RMSD is calculated based on atom selection”. What is this atom selection?

10. The discussion about RMSD from protein and Ligand is not clear, and What do authors mean when they say that “The RMSD of ligand is in agreement with protein movement with an RMSD of 7.5 angstrom.” Authors never discussed the conformations and orientations of the ligand through the 100 ns simulation. Figures of at different times must be provided.

11. Figure 3 must be improved and well described. For example, Fig 3A has two RMSD plots and is not written what is the difference. Figure 3B there are some green lines, What do they mean? Figure 3C, What do the different colors mean? Etc.

12. Regarding the methodology, there is missing crucial information. For example, How they perform the minimization? Authors said “structure minimization for less than 0.3”, What does it mean? Why do authors use 3.975 mM ion strength, usually biological simulations are performed at 0.15 mM, barostat, thermostat, integrator, short range interaction model, etc. is missing in MD methods.

13. Also, a relaxation protocol and minimization of the system must be done before MD. And perform the simulation with at least one replica.

14. Line 469, section 4.9 authors said, “basic set” What is a basic set? It does not exist and through the manuscript there are many conceptual mistakes like this.

Finally, in the discussion section, authors never discussed the results. It appears more as an introduction.

Author Response

We thank the reviewer and have addressed the reviewer comments/suggstions and please see the attachment for the detailed response

Reviewer 3 Report

Manuscript Number: cimb-2300185

Title: Denovo design of anti-COVID drugs using machine learning based equivariate diffusion model targeting the spike protein

By V. Niranjan еt al.

The work has some advantages, but from my point of view, it requires major corrections before publishing.

Here are some suggestions:

1.         In the Introduction part, the significance, purpose, and importance of this investigation cannot be found.

It presents general information on "Diffusion models for molecules" and "Structure-based drug design", but this information should be linked to the need for this research and the main conclusions of the work performed.

2.         Please, give the structure of the compounds docked in Table 2 instead of Compound name, for clarity.

3.         In the Discussion part lack of discussion and interpretation of own results. Instead, the authors present known information about the potential of the Heptad repeat domain (HRD) on the spike protein of SARS-CoV-2 a drug target and as well as known information about the advantages and limitations of the machine learning-based diffusion models in the development of new drugs.

4.         Please, check the References carefully. The abbreviated journal names miss in a lot of places. The ref. [21] is not relevant to the research.

5.         Many of the meanings of the abbreviations used are not given when the abbreviation is first mentioned in the text.

Author Response

(The authors gave the same response as above.)

Round 2

Reviewer 1 Report

Thanks for your etensive reformatting, and for accepting most of my observations - including a better explanation of some potential issues, which often dispelled my concerns.

I do believe that additional efforts could further improve the manuscript, but I also state here that its quality and clarity has been significantly improved, and that it could be accepted after minor revisions.

Author Response

We thank the reviewer for accepting the manuscript. As the suggestions from other reviewers the manuscript has been improved both in its scientific merit and english language.

We thank you again for your time and effort.

Reviewer 2 Report

The manuscript entitled “Denovo design of antiCOVID drugs using machine learning-based equivariate diffusion model targeting the spike protein” relies on computational studies of new compounds for COVID-19 treatment. Although, authors made some changes to improve the manuscript there are still being conceptual mistakes and the results are still poor presented, machine learning methodology and results found are not well described (as tittle suggests) and the discussion is still missing it appears more as an introduction. Results found are not discussed in this section, authors did not make any change to it.

In my view there are also phrases that I cannot understand, as for example:

Using the ML models 196 compounds were denovo compounds were generated which had no hits on any major chemical databases.”

compounds were denovo compounds were generated… there is a grammatical mistake, and de novo is written by two letters.

The introduction is still poor presented and there are missing references, for instance:

Several approaches have been explored for designing drugs that target the heptad domain of the spike protein. One strategy is to develop small molecule inhibitors that interfere with the interaction between HR1 and HR2, thus preventing the formation of the six-helix bundle and inhibiting virus entry into host cells. Another approach is to develop peptides or antibodies that mimic the structure of HR1 or HR2 and can bind to the com- plementary heptad region, disrupting the fusion process.”

There are not references in this information.

I ask authors to homogenize the text when they talk about COVID-19, because every paragraph they named it different or with a different naming convention. But they did not change this small modification.

Regarding the figures, in the revised version they are illegible. I cannot see almost nothing due to the figure quality.

In molecular docking results. Authors stated:

The compounds once clustered were subjected to molecular docking. The compounds from the inpainting model 10/98 compounds were docked showing energy value > -2kcal/mol. The rest of the compounds had a positive dock score which is not thermodynamically viable binding energy. “

I think there is a mistake, because values are lower than -2kcal/mol (as I can see in Table 2) not higher than -2kcal/mol.

In accordance with MD simulations, authors explained that they performed the replica but only reported the RMSD of both calculations. But, Figure 3B, C, D and E is exactly the same as in the old version. I think they did not find exactly the same results in both replicas and cannot find these data either in the main manuscript or the supplementary information.

What was the reason for using 303 K and not 310 K for the MD simulation?

For small globular proteins it is expected to be in the range of 3 to 6 angstroms.” Missing reference.

I thank the authors for reporting the RMS of the ligand for different frames, but I suggest to move Table 5 to the SI and just analyze it in the text. And could be better to put the figure of the whole protein and the different frames in order to see if the ligand remains in the docked position or move to another region of the protein. And discuss it.

Regarding MMGBSA calculations if authors perform the MD replica. Why they did not perform also the MMGBSA for this new calculation. I saw that they have the same energy values as in the previous version.

According to my question about a “basic set”, I disagree with the author’s response. It is true, all the description about 6-31G but it is a basis set not a basic set. I think there are still conceptual mistakes in the manuscript.

Author Response

We thank the reviewer for this time and effort. All the suggestions and queries have been addresed in the revised manuscript.

Please find attachedment the response letter for the queries.

Reviewer 3 Report

Manuscript Number: cimb-2300185

Title: Denovo design of anti-COVID drugs using machine learning-based equivariate diffusion model targeting the spike protein

By V. Niranjan еt al.

Following the corrections made by the authors, I consider that the paper is suitable for acceptance for publication in Current Issues in Molecular Biology in the present form.

Author Response

We thank the reviewer for the time and effort. Our heartfelt thanks for accepting the manuscript after revisions.